# Deadwood Amount at Disturbance Plots after Sanitary Felling

**DOI:** 10.3390/plants11070987

**Published:** 2022-04-05

**Authors:** Ján Merganič, Katarína Merganičová, Mária Vlčková, Zuzana Dudáková, Michal Ferenčík, Martin Mokroš, Vladimír Juško, Michal Allman, Daniel Tomčík

**Affiliations:** 1Department of Forest Harvesting, Logistics and Ameliorations, Faculty of Forestry, Technical University in Zvolen, T.G. Masaryka 24, 96001 Zvolen, Slovakia; vlckova@tuzvo.sk (M.V.); zuzana.allmanova@gmail.com (Z.D.); ferencik@tuzvo.sk (M.F.); martin.mokros@gmail.com (M.M.); jusko@tuzvo.sk (V.J.); allman.michal@gmail.com (M.A.); dtomcik4@gmail.com (D.T.); 2Faculty of Forestry and Wood Sciences, Czech University of Life Sciences Prague, Kamýcká 129, 6-Suchdol, 16500 Praha, Czech Republic; k.merganicova@forim.sk; 3Department of Biodiversity of Ecosystems and Landscape, Institute of Landscape Ecology, Slovak Academy of Sciences, Akademická 2, 94901 Nitra, Slovakia

**Keywords:** fine woody debris, volume, coverage of deadwood, photogrammetry, model, impact of site, accuracy and precision

## Abstract

Deadwood is an important component of forests that fulfils many ecosystem functions. The occurrence, amount and spatial distribution of deadwood in forest ecosystems depend on tree species composition, historical development and past management. In this presented study, we assessed the total amount of deadwood, including fine and coarse woody debris at five areas of predominantly broadleaved forests within the University Forest Enterprise of the Technical University in Zvolen, Slovakia that had been disturbed by windstorm Žofia in 2014. Windthrown wood was salvaged between May 2014 and October 2015. In the year 2018, we performed an inventory of deadwood that remained on-site after salvage logging. The mean volume of deadwood recorded at sample plots fluctuated between 35.96 m^3^/ha and 176.06 m^3^/ha and mean deadwood coverage values at individual disturbed areas ranged from 7.27 to 17.91%. In the work, we derived several models for the estimation of deadwood volume based on deadwood coverage and/or diameter, which showed that these characteristics are good proxies of deadwood volume. The tests, involving close-range photogrammetry methods for deadwood quantification, revealed that the number of pieces and the coverage of deadwood recorded in photos was significantly lower than the values derived from field measurements.

## 1. Introduction

Deadwood is an important component of forests that fulfils a number of ecosystem functions. The term deadwood encompasses all woody material that occurs in an ecosystem as a result of tree mortality caused by natural or human-induced disturbances, including standing or fallen dead trees, and their parts [1,2,3]. It is a valuable habitat and/or food source for many living organisms, and hence has a positive impact on forest biodiversity [2,4,5,6,7,8]. Due to this, it is used as one of the biodiversity indicators [9]. Deadwood is a long-term source of nutrients in forest ecosystems [2,10], particularly calcium and magnesium [11]. It also represents substantial carbon storage [12,13]. In mountainous areas, deadwood provides suitable conditions for natural regeneration [14,15,16,17,18,19,20,21]. Deadwood lying on the ground protects soil from water or wind erosion [22,23] and prevents rockfalls and avalanches in mountainous areas [24]. It affects the forest microclimate and serves as water storage during dry periods in a year [2]. It also has a positive impact on forest productivity [25].

The occurrence, amount and spatial distribution of deadwood in forest ecosystems depend on tree species composition, historical development and past management. Managed forests usually contain less deadwood than old-growth forests left for self-development [26,27,28,29,30,31]. The amount of deadwood in broadleaved forests is often smaller than in coniferous ones because their productivity is lower, and they are usually less damaged by abiotic factors [32]. Nevertheless, with the ongoing climate change, disturbance events have become more frequent in the whole Central Europe regardless of tree species [33]. Recently, disturbances have been driving forest management interventions that focus on the processing of wood damaged by disturbances [34]. In spite of sanitary fellings applied in forests after disturbances, a certain amount of deadwood always remains on-site. The remaining volume depends on a number of conditions, including site (terrain, slope, accessibility), and wood characteristics (dimensions, damage, spatial distribution) [26]. Although the amount of fine woody debris at disturbed sites may be considerable [35], it is often omitted from the studies because of its small dimensions. Therefore, data about fine deadwood is scarce, although its importance as a habitat for multiple deadwood-associated fungal species is known [36,37]. Omitting fine woody debris from surveys may substantially underestimate the information on the richness and abundance of fungi [36,37]. Hence, more information about fine deadwood is needed from multiple forest ecosystems to provide a complete picture of the state of woody debris in forests.

In the presented study, we would like to fill this gap by assessing the total amount of deadwood, including fine and coarse woody debris in areas of broadleaved forests that had been disturbed by the wind. The aim of the study was to provide comprehensive information about the deadwood in windthrown areas after salvage logging had been applied. Hence, we hypothesised that the amount of remaining deadwood, especially coarse woody debris, would be low. Due to this, we analysed fine and coarse woody debris separately. Furthermore, since measuring individual pieces of deadwood, especially fine woody debris during field inventories is laborious and time-consuming, we also examined other possibilities for estimating the volume of present deadwood using (1) the information on relative deadwood coverage in disturbed areas, and (2) modern remote sensing methods that could speed up the collection of data about deadwood. This is particularly true if fine woody debris is to be included because the dimensions of fine deadwood pieces are small. Therefore, we tested the applicability of the above-mentioned approaches to estimate deadwood volume. We also examined the impact of site conditions on the amount and coverage of deadwood to reveal their main drivers at sites with active post-disturbance management.

## 2. Results

### 2.1. State and Amount of Deadwood at Disturbed Areas

During the on-site inventory of woody debris at sample plots of all disturbed areas, we recorded 8510 deadwood pieces in total, out of which 8387 pieces originated from broadleaved trees and the remaining 123 pieces were coniferous. The majority of pieces (8106 pcs, i.e., 95%) were categorised as fine woody debris. The mean diameters of deadwood pieces varied from 18.6 mm at disturbed area No. 4, up to 28.2 mm at area No. 5.

From the point of wood decay, the major proportion of deadwood pieces (42.3%) and of deadwood volume (72.7%) belonged to the first, i.e., the least decomposed class. About one-third (27.4%) of all deadwood pieces or 14% of the deadwood volume belonged to the second decay class, i.e., the wood was moderately soft. Similarly, one-third of the deadwood pieces (30.3%) and 13.3% of the deadwood volume were in the third decay class, i.e., the wood was soft and easy to defragment.

The analysis revealed that the mean volume of deadwood recorded at sample plots per unit area fluctuated between 35.96 m^3^/ha and 176.06 m^3^/ha depending on the disturbed area, while the volume of coarse and fine woody debris varied from 21.82 to 145.86 m^3^/ha and from 14.15 m^3^/ha to 30.2 m^3^/ha, respectively (Figure 1). The highest deadwood volume per hectare was recorded at disturbed area No. 5, and the lowest volume at area No. 1. The same results were found also for coarse and fine woody debris separately.

The total volume of deadwood estimated from the inventory data varied from 35 m^3^ to 2217.3 m^3^ depending on the disturbed area, while the volume of fine woody debris fluctuated between 16.8 m^3^ and 380.3 m^3^ (Table 1). The results in Table 1 demonstrate that with 95% confidence, the actual total deadwood volume at disturbed area No. 5 was in the range of 1427 to 3007 m^3^. This amount is significant from an ecological point of view.

### 2.2. Deadwood Coverage

Deadwood coverage is an interesting variable from an ecological point of view. Our results showed that mean deadwood coverage values in individual disturbed areas ranged from 7.27% to 17.91%. Coarse woody debris covered on average between 2.5% and 6.91% of the sample plot area, while fine woody debris covered from 4.77% to 11% of the area (Figure 2). Similarly, as in the case of volume, the highest coverage was observed at disturbed area No. 5, and the lowest at area No. 1.

### 2.3. Impact of Site Characteristics on Deadwood Amount and Coverage

Next, we analysed relationships between deadwood volume or coverage and site characteristics using univariate correlation analyses (Table 2). The most significant correlation of deadwood volume was found with the pattern of its spatial distribution at a plot. A higher volume of deadwood was associated with its regular distribution at the plot. Deadwood volume also significantly increased with the increasing slope. Negative significant correlations were revealed between deadwood volume and the coverage of vegetation or plant litter, i.e., deadwood volume decreased with the increasing vegetation or plant litter coverage. Some microrelief forms also significantly correlated with deadwood volume: convex forms of microrelief along the slope and the contour line had a positive impact on deadwood volume, while the influence of flat terrain was negative.

Deadwood coverage was most significantly correlated to the number of deadwood pieces, followed by the regular pattern of spatial deadwood distribution at a plot. Both characteristics had a positive impact on deadwood coverage, while vegetation coverage had the most significant negative impact on deadwood coverage. The spatial pattern of vegetation distribution also significantly affected deadwood coverage (Table 2).

In the case of fine woody debris only, the closest significant correlations of volume or coverage were revealed with the number of deadwood pieces. Vegetation coverage and distribution also had significant impacts, as in the case of the total deadwood coverage. However, some relationships changed their character. For example, the volume of fine woody debris decreased with the increasing slope but increased with the increasing coverage of plant litter (Table 2).

### 2.4. Models of Deadwood Volume

We derived several models for estimating deadwood volume based on deadwood coverage and/or log diameter (Table 3) and examined their quality using several statistics methods (R^2^, F-statistics, AIC, BIC). As the results in Table 3 show, the best model was the polynomial model of the second degree with two predictors: deadwood coverage and mean diameter of deadwood piece, which explained 91 % of the variability in deadwood volume.

The polynomial model (No. 6 from Table 4) has the following form:Deadwood volume (m^3^/ha) = −5.227836622 − 6.406274126 × Coverage + 0.158176531 × Coverage ^2 + 1.545334215 × Diameter + 0.18518785 × (Coverage × Diameter) − 0.009702535 × Diameter ^2(1)
where coverage represents the deadwood coverage of the area in %, and diameter in mm is the mean diameter of all deadwood pieces in the area. This model gives the total volume of aboveground deadwood at a plot that includes coarse and fine woody debris. Statistical characteristics of the model are presented in Table 4.

We performed a similar analysis for the volume of fine woody debris (FWD). Statistical characteristics of derived models presented in Table 5 suggest that nonlinear models No. 4, 6, and 7 were most suitable since they had the lowest values of AIC and BIC. The best one was model No. 7, which was derived from model No. 4 by excluding the diameter as a predictor because of its insignificant impact.

Statistical characteristics of model No. 7 are presented in Table 6. Its final form is as follows:Volume of fine woody debris (m^3^/ha) = −1.59631449 + 1.24054005 × Coverage + 0.07015479 × Coverage × Diameter(2)
where Coverage represents the coverage of FWD at a plot in %, and Diameter represents the mean diameter of FWD in mm.

Figure 3a presents the relationships between the deadwood volume (m^3^/ha) and its coverage (%) described by three derived models, the best model (No. 6 in Table 3), a simple linear regression (model No. 1 in Table 3), and a simple nonlinear regression (model No. 5 in Table 3). The comparison of the latter two models on the base of their statistical parameters (Table 3) indicates that the nonlinear model described the relationship better. The same simple linear and nonlinear models of FWD volume depending on FWD coverage are presented in Figure 3b. Their development indicates that the relationship had a linear trend, particularly if FWD coverage was less than 50%. The red line in Figure 3b presents the relationship of FWD volume to the coverage described by the best model (model 7 in Table 5).

### 2.5. Accuracy and Precision of Deadwood Quantification Using a Photogrammetric Method

The results of the analysis showed that the deadwood quantification in orthophoto images was not exhaustive, since both the number of pieces (F = 12.29 ***) and the deadwood coverage (F = 11.4 ***) recorded in photos were significantly lower than the values derived from field measurements. Significant differences were revealed for both variables and all disturbance areas (Figure 4). The differences between the deadwood quantification in orthophotos and field measurements tended to increase with the increasing level of difficulty to identify deadwood (Figure 4a,b). However, a negative significant impact of the identification difficulty on the assessment of the number of deadwood pieces or deadwood coverage was observed only at disturbance area No. 2, where the difference significantly increased with the increasing difficulty to identify deadwood. Although mean differences indicate the underestimation of both the number of deadwood pieces and the deadwood coverage, individual differences in the number of pieces were both positive and negative, while most values of deadwood coverage estimated from images were smaller than those measured in the field.

The relationship between the number of deadwood pieces recorded from the images and in the field (Figure 5a) showed that if less than 27 deadwood pieces occurred at a plot, the evaluators had a tendency to underestimate their number. If this threshold was exceeded, the evaluators overestimated the number of deadwood pieces. On average, six deadwood pieces per plot were not identified during the image evaluation. The estimation of the number of deadwood pieces from orthophoto images was significantly affected by evaluators (F = 16.4 ***). The mean number of deadwood pieces identified in the images did not significantly differ from the values observed in the field, only in the case of two evaluators (person No. 3 and 4, Figure 5a).

Deadwood coverage derived from images was systematically underestimated across the whole observed range regardless of the evaluators, while the bias slightly increased with the increasing coverage (Figure 5b). The average deadwood coverage derived from images was 7% lower than the coverage determined in the field. The statistical analysis did not reveal any significant differences in the deadwood coverage estimation between the evaluators (F = 0.643).

## 3. Discussion

Deadwood is one of the most important components of forest ecosystems [38], storing a large amount of carbon [12,13]. A deadwood amount is often used as an indicator of sustainable forest management [39,40]. Several studies found that species diversity depends on the volume of deadwood that is present in a forest [41,42]. It has been reported that the deadwood amount of 20–30 m^3^/ha can provide a suitable habitat for a wide range of species [8,25,43,44]. Our results revealed that despite active forest management being applied in the region of interest, the deadwood volume after sanitary felling was not negligible since it varied between 36 and 176 m^3^/ha. This is in accordance with the mean values of the deadwood amount reported from multiple European forests that fluctuated from 40 to 200 m^3^/ha [45,46,47,48,49]. Similar values (0–258 m^3^/ha) were obtained from ICP Forests monitoring, whereas the highest proportion of ICP plots contained less than 50 m^3^/ha [50]. Deadwood volume of more than 100 m^3^/ha was observed in mountainous areas of Central Europe, while forest ecosystems of Great Britain and the Mediterranean region contained less deadwood [50]. The amount of deadwood exceeding 400 m^3^/ha was recorded in old-growth forests of Central Europe, e.g., in Slovakia [51], Poland [52], or Slovenia [53].

The share of fine and coarse woody debris depends on the forest type [38]. In our study, the volume of coarse woody debris varied from 21.82 to 145.86 m^3^/ha, representing 50% to 80% of the total deadwood quantity. The absolute CWD values at two disturbance areas (Figure 1) corresponded with the volumes obtained from Slovak national monitoring, reported for the oak-beech vegetation zone (24–28 m^3^/ha, [54]), to which all investigated disturbed areas belong. The volumes at three other areas exceeded these values by two to six-folds, indicating that a large amount of wood remained at sites after salvage logging. In addition, the results showed that one fifth to one half of the total deadwood volume occurred at sites in the form of fine woody debris (14 to 30 m^3^/ha, or 20% to 50% of the total deadwood volume). This suggests that FWD is not a negligible pool of forest ecosystems. Moreover, it is an important habitat for many fungal species that may be overlooked if FWD under certain thresholds is not inventoried [37]. This may have subsequent implications for biodiversity studies.

The majority of works dealing with deadwood focus on its amount. However, deadwood coverage is another important factor affecting different processes in ecosystems. Lying deadwood increases surface roughness, which prevents runoff from increasing its volume and speed [23,55]. Deadwood biomass on the ground also protects soil from the impact of wind and/or water [56]. In Utah and Montana, in the U.S.A., the reduction of soil cover by plant biomass from 100% to 1% caused an increase in erosion intensity by 200 times [57]. Moreover, significantly higher soil losses were observed on skidding trails with a lack of deadwood and plant cover in comparison to reference or harvested plots [22]. Soil erosion, in combination with other disturbances, may further reduce forest sustainability and soil fertility [58].

The authors in [59] evaluated the impact of CWD coverage on the development of recultivated ecosystems (oil sands land) from the point of erosion, diversity and the storage of organic matter over a longer term. They found that a low coverage of woody debris (up to 30 % ground cover) seems to be optimal for maintaining native plant species diversity and abundance while controlling undesirable plant species. The authors in [60] used CWD coverage for modelling the temporal and spatial dynamics affected by various clearcutting and fire regimes over a 1000-yr period. The coverage of WD was also evaluated in studies dealing with the activities of small mammals [61,62]. Greater coverage of CWD increased the occurrence of red-backed voles (*Clethrionomys gapperi*) [62]. The authors in [61] recommended coverage of downed CWD between 15 and 20 percent for favouring communities of small animals. Deadwood coverage values, which were found in our study were either below or at the bottom threshold of the suggested range (mean deadwood coverage values at individual disturbed areas ranged from 7.27% to 17.91%, while coarse woody debris covered between 2.5% and 6.91% of the sample plot area, and fine woody debris covered from 4.77% to 11% (Figure 2)). However, as [62] pointed out, the threshold values of biomass amount depend on the management goal. While diversity studies aim at increasing deadwood, fuel management tends to reduce CWD in forests to minimise the risk of wildfire occurrence and/or intensity. The authors in [63] analyse the relationship of CWD coverage to the fire risk. Their models, which described the relationship of volume to coverage revealed a linear relationship, similarly to our model for fine woody debris. The tight relationship of woody debris coverage to volume suggests that this characteristic can be used as an alternative ecological variable to volume.

Remote sensing technologies seem to be promising tools for deadwood inventory. Infrared aerial photographs are suitable for mapping and quantification of standing deadwood [64,65]. Aerial laser scanning can be used for the preliminary mapping of forests with a great amount of lying woody debris [66,67,68]. The majority of studies quantify coarse woody debris. However, recent developments in LiDAR technology suggest that this technology has the capacity to also capture finer details. Hence, in future, it may be used in applications for ecological topics and forest management. Newly available platforms (e.g., mobile laser scanner) and sensors (e.g., multispectral laser scanner) might also provide new opportunities for this field [69]. Furthermore, complex forest structures and dense understory vegetation challenge point cloud processing tools and may cause errors in measurements. The authors in [70] applied deep learning for semantically segmenting high-resolution forest point clouds from multiple different sensing systems under diverse forest conditions. Their work showed that the accuracies of CWD segmentation were lowest, at the level of approximately 55%. Our results confirmed that WD identification with these technologies is still difficult. The coefficients of variation for the number of deadwood pieces and deadwood coverage were 225% and 114%, respectively; such a high variability of variables requires high sampling intensity. Moreover, the values obtained from orthophoto images were, in most cases, negatively biased, which is mainly caused by the overlapping of objects. Hence, the approaches need further developments and additional improvements to increase the identification accuracy.

## 4. Materials and Methods

The inventory was performed at a part of the University Forest Enterprise of the Technical University in Zvolen (UFE) situated in the middle of Slovakia, central Europe (48°37’51.42’’ N 19°2’38.83’’ E). UFE is a specialised centre at the Technical University in Zvolen used for students’ training. It manages almost 10,000 ha of forests that occur at elevations from 250–1026 m a.s.l. and covers varying soil, climatic and vegetation conditions that are suitable for research activities. The mean annual air temperature varies from 4 to 8 °C, and the annual precipitation total varies between 600 and 1000 mm, depending on the elevation. The enterprise is located in the volcanic mountain range of the Kremnické Mts. and Javorie Mts., hence, bedrocks of volcanic origin prevail. Cambisols are the most common soil types in the enterprise.

Forests are composed of 27 different tree species, predominantly broadleaved ones covering 85% of the total enterprise area. The European beech (*Fagus sylvatica* L.) is the dominant species (51%), followed by oak spp. (*Quercus* spp.-16%) and hornbeam (*Carpinus betulus* L.-8%). Of the coniferous tree species, Norway spruce (*Picea abies* L.) is the most common one (7.8%) followed by silver fir (*Abies alba* Mill.-3.2%), and pine (*Pinus sylvestris* L.-2.7%). Active forest management is performed in the majority of enterprise forests (86%).

The empirical material for the analyses was obtained from the statistical inventory of deadwood performed at 5 windthrown and subsequently salvaged areas in the year 2018. The areas were destroyed by windstorm Žofia on 14 and 15 May 2014. During the event, wind speeds reached 100 km/h in gusts. The windstorm damaged all trees within the identified areas. Sanitary logging was performed within one and half years after the disturbance between May 2014 and October 2015, during which most of the damaged wood was extracted from the parts that were disturbed. We selected the disturbed areas because the visual survey prior to the inventory indicated differences in the amount and spatial distribution of deadwood between them. Moreover, the areas were of different sizes and shapes (Figure 6), within which the salvage logging typical for actively managed forests was applied after the disturbance, and prior to the disturbance, they were covered by mature forest stands (Table 7) that are usually most prone to wind disturbances. In addition, the areas have been a subject of our research interests, particularly from the point of applying different methods of remote sensing for a longer time [71,72]. The location and basic characteristics of the areas are presented in Figure 6 and Table 7.

In April and May 2018, i.e., four years after the windthrow event and two and half years after the end of the salvage logging, we established 536 sample plots at all the disturbed areas in total, to examine the on-site conditions of the deadwood amount remaining after the sanitary logging of merchantable wood was finished. Each sample plot was of a square shape with a side length of 1 m, i.e., its area was 1 m^2^. The applied sampling system can be characterised as a randomised systematic sampling. Prior to the fieldwork, each disturbed area was systematically divided, and the positions of the sample plots were generated within individual grid cells. In the field, the crew navigated themselves to the generated points with GPS Garmin 60CSx. The side of the sample plot from the bottom left corner was oriented towards the north.

The information spectrum consisted of 17 characteristics describing site conditions (e.g., slope, aspect, micro-relief (- flat land, ^ concave terrain, v convex terrain) along the contour line and along the slope, i.e., 9 possible combinations were identified in total, e.g., -^ is flat land along the contour line and concave terrain along the slope), vegetation coverage, the spatial distribution of vegetation, deadwood coverage and the spatial distribution, form and coverage of humus, etc., and 5 variables specifying individual deadwood pieces (tree species category (coniferous, broadleaved), length, diameter, degree of decay, spatial orientation of the deadwood piece). All characteristics of the information spectrum were assessed at the plot. All deadwood pieces with a minimum length or height of 5 cm and a diameter at the top end of at least 0.5 cm were measured. Deadwood lengths and heights were measured with a measuring tape with an accuracy of 1 mm. Deadwood diameters were measured with a calliper, also with an accuracy of 1 mm. At least three different diameters were determined for each deadwood piece (bottom, middle and top diameter). Each deadwood piece was assigned a decay stage using three decay classes: (1) fresh, solid wood, (2) moderately soft deadwood, and (3) very soft deadwood that defragments easily. The threshold between fine and coarse deadwood was set to a diameter of 7 cm at the top end.

The volume of each deadwood piece was calculated using the Huber formula based on the quadratic mean of at least three diameters (top, middle, bottom) and the length of the piece [73,74]. Volumes of all deadwood pieces were summed up to obtain the plot-specific deadwood volume, which was subsequently multiplied by an expansion factor of 1000 to convert it to the area of 1 hectare.

Deadwood coverage was defined as a proportion of a plot area covered by all occurring deadwood pieces placed next to each other on the ground. The coverage of each deadwood piece was calculated as an area of a rectangle with sides equal to the quadratic mean diameter and the length of the piece. The deadwood coverage at a plot was obtained as a sum of the coverage values of all deadwood pieces divided by the total plot area (i.e., 1m^2^).

The actual condition at each sample plot was also captured with a digital camera Panasonic DMC FZ1000 (1” MOS sensor, 20,1 Mpx, produced by Panasonic, Amsterdam, Netherland) using the methods of terrestrial photogrammetry. Digital images of 124 sample plots were processed with the Agisoft Photoscan Professional software (Agisoft LLC, St. Petersburg, Russia) creating their orthophoto images (Figure 7). The images were subsequently analysed by 5 independent evaluators to test the precision and the accuracy of the photogrammetric method for quantifying deadwood consisting primarily of fine woody debris. Prior to the assessment, the selected sample plots were divided into three categories of difficulty to identify deadwood (easy, moderately difficult, and difficult to identify represented by 49, 50, and 25 plots, respectively) according to the number of deadwood pieces at a plot (easy: from 0 to 15pcs; moderately difficult: containing from 16 to 35 pcs; and difficult: with more than 35 pieces at a plot). Vectorisation of deadwood at the sample plots was performed in QGIS by each evaluator individually. All visually identified deadwood pieces by the respective evaluator inside the sample plots with a minimum length or height of 5 cm and a minimum top diameter of 0.5 cm were vectorised. Each deadwood piece was represented by an individual polygon. This information was used to derive the coverage of deadwood, defined as the proportion of the sample plot area covered by all deadwood pieces if they were placed next to each other. For the mathematical-statistical processing of data, we used standard statistical approaches (Student’s t-test, univariate and multivariate correlation and regression analyses, ANOVA) using the available tools in the R environment [75]. Categorical variables (e.g., microrelief, a form of humus, the spatial distribution of deadwood) were transformed into dummy variables. The ggplot2 package was used to visualise the results [76].

## 5. Conclusions

The presented study brings new information about deadwood remaining in managed temperate Central European forests based on the analyses of the situation at five big, disturbed and subsequently salvaged areas (0.75 to 12.59 ha per area, in total 25.38ha) destroyed by wind Žofia in 2014. The results revealed that the remaining volume of deadwood three years after salvage logging was substantial and fluctuated between 35.96 m^3^/ha and 176.06 m^3^/ha, depending on the disturbed area. The volume of coarse and fine woody debris varied from 21.82 to 145.86 m^3^/ha and from 14.15 m^3^/ha to 30.2 m^3^/ha, respectively. This suggests that FWD is not a negligible pool of forest ecosystems and should not be overlooked in inventories, as it may have subsequent implications for biodiversity studies. The presented values provide valuable insights into how much deadwood remains at large-scale disturbed sites after salvage logging and may be used as proxies (average from all disturbance areas with a 95% error is for all woody debris-124.13 ± 32.7 m^3^/ha, coarse woody debris-98.29 ± 32.6 m^3^/ha, fine woody debris -25.83 ± 2.2 m^3^/ha) for estimating the situation in similar beech-dominated forests of the Carpathians, where active post-disturbance management is applied.

Our results demonstrated that instead of measuring the dimensions of each WD piece, deadwood coverage may be estimated and used to derive its volume, as the correlation between these two variables was high. We derived several models for the deadwood volume estimation based on deadwood coverage and/or mean deadwood diameter. The best model was the polynomial model of the second degree with two predictors: deadwood coverage and the mean diameter of deadwood, which explained 91% of the variability in deadwood volume. Nonlinear models (multiple regressions with interactions of independent variables) with coverage and diameter as the predictors of deadwood volume were also derived separately for fine woody debris.

The deadwood amount significantly increased with the increasing slope, and convex forms of microrelief along the slope and contour line, while it decreased with the increasing vegetation or plant litter coverage, and flat terrain. This indicates that terrain conditions affected post-disturbance sanitary logging. The analysis of the photogrammetry methods (low-cost technology with a classical camera) for deadwood quantification revealed that the number of pieces and the coverage of deadwood recorded in photos was significantly lower than the values derived from the field measurements. Hence, further improvements are needed to apply these modern methods to deadwood inventory.

## Figures and Tables

**Figure 1 plants-11-00987-f001:**
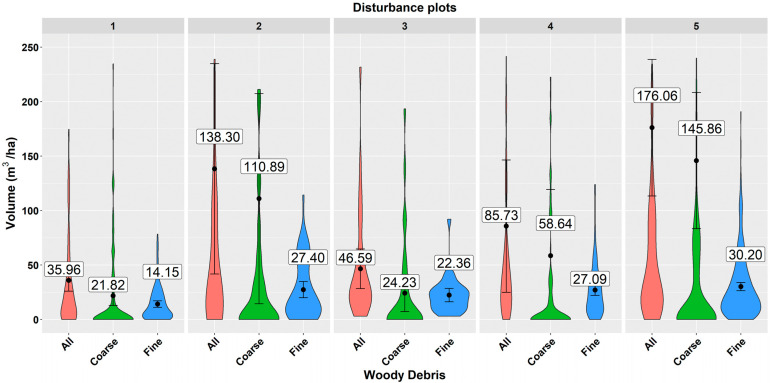
Deadwood volume at individual disturbed areas. A violin plot depicts distributions of numeric data using density curves. The width of each curve corresponds with the approximate frequency of data points in each region. The black point and the value represent the mean deadwood volume and the error bar shows a 95% confidence interval of the mean.

**Figure 2 plants-11-00987-f002:**
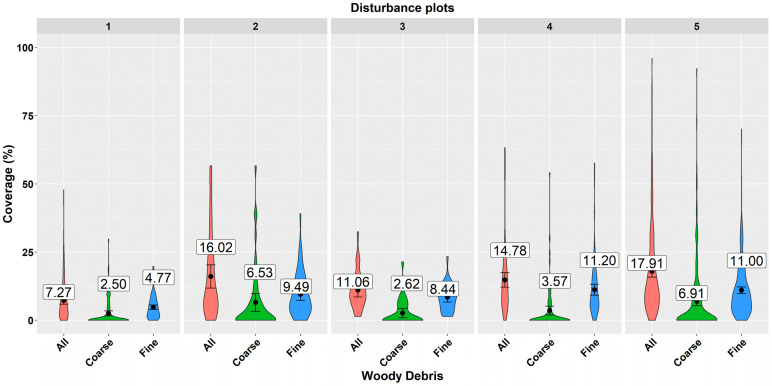
Deadwood coverage frequency at individual disturbed areas. A violin plot depicts distributions of numeric data using density curves. The width of each curve corresponds with the approximate frequency of data points in each region. The black point and the value represent the mean deadwood coverage and the error bar shows a 95% confidence interval of the mean.

**Figure 3 plants-11-00987-f003:**
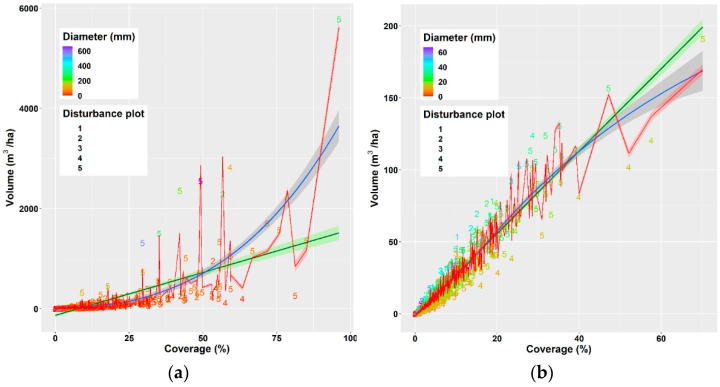
Relationships between volume of deadwood (**a**), volume of fine woody debris (**b**) and its coverage are described with a simple linear regression (green lines), a nonlinear regression (blue lines) or nonlinear models with coverage and diameter as predictors (red lines). Positions of numbers referring to individual disturbance areas in graphs indicate measured values of deadwood coverage and volume and their colours indicate the values of respective mean deadwood diameter.

**Figure 4 plants-11-00987-f004:**
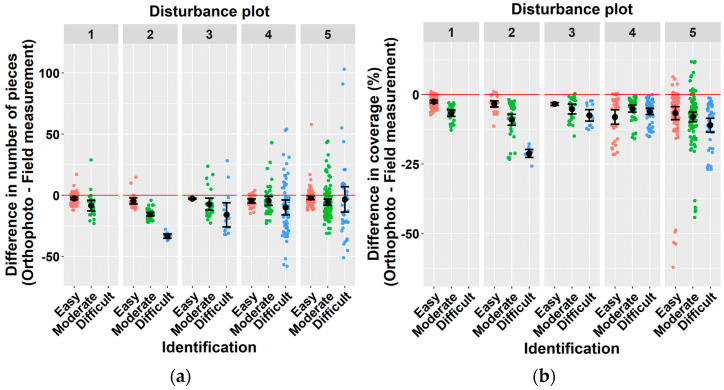
Impact of the difficulty of deadwood identification on the accuracy of number of deadwood pieces (**a**) and deadwood coverage (**b**) estimated from orthophoto images. Identification difficulty was specified based on the number of deadwood pieces on the plot (easy = 0–15 pcs, moderate = 16–35 pcs, difficult= more than 35 pcs of deadwood).

**Figure 5 plants-11-00987-f005:**
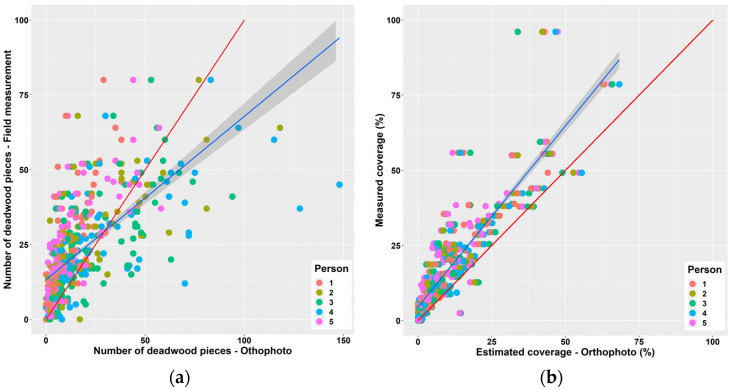
Relationship between estimated (orthophoto images) and the observed number of deadwood pieces (**a**) and coverage (**b**). The blue line shows the linear regression between estimated and observed values regardless of evaluators (persons). The red line is a reference 1:1 line.

**Figure 6 plants-11-00987-f006:**
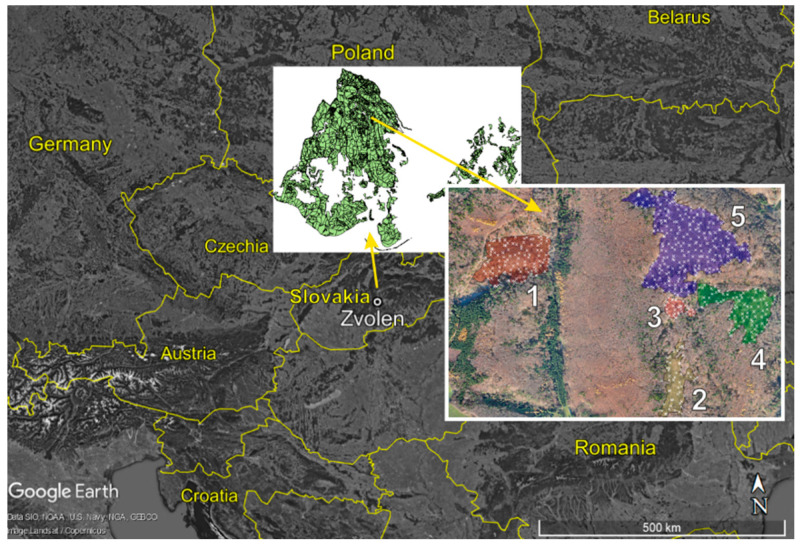
Location of disturbed areas within University Forest Enterprise of Technical University in TA \l “Mokroš et al. 2017” \s “Mokroš et al. 2017” \c 1 Zvolen, Slovakia.

**Figure 7 plants-11-00987-f007:**
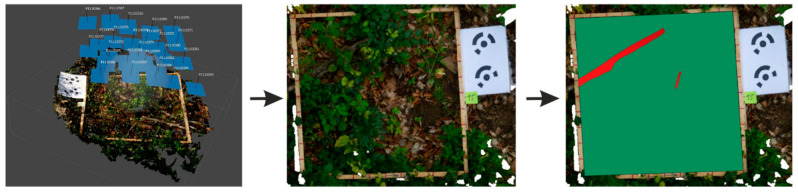
Deriving of orthophoto images of sample plots in Agisoft Photoscan Prof. and polygonisation of deadwood in the orthophoto image.

**Table 1 plants-11-00987-t001:** Total volume of deadwood in disturbed areas based on inventory results.

Area	Deadwood	*n*	Total Volume	SD	SE	95 %IS	95 %DI	95 %HI	Proportion
			(m^3^)						
1	All	110	190.1	277.7	26.5	52.5	137.6	242.6	100.0
	Coarse	110	115.3	246.7	23.5	46.6	68.7	161.9	60.7
	Fine	110	74.8	83.2	7.9	15.7	59	90.5	39.3
2	All	52	337.8	846.6	117.4	235.7	102.1	573.5	100.0
	Coarse	52	270.9	846.1	117.3	235.6	35.3	506.4	80.2
	Fine	52	66.9	65.6	9.1	18.3	48.7	85.2	19.8
3	All	31	35	37.2	6.7	13.6	21.3	48.6	100.0
	Coarse	31	18.2	34.8	6.3	12.8	5.4	30.9	52.0
	Fine	31	16.8	12.6	2.3	4.6	12.2	21.4	48.0
4	All	93	368.9	1268.9	131.6	261.3	107.6	630.2	100.0
	Coarse	93	252.3	1269.9	131.7	261.5	−9.2	513.9	68.4
	Fine	93	116.6	106	11	21.8	94.7	138.4	31.6
5	All	251	2217.3	6353.7	401	789.8	1427.5	3007.2	100.0
	Coarse	251	1837	6334.3	399.8	787.4	1049.6	2624.5	82.8
	Fine	251	380.3	377.3	23.8	46.9	333.4	427.2	17.2

Legend: *n*—number of sample plots, SD—standard deviation, SE—standard error, 95 %IS—error of total deadwood volume at 95 % confidence interval, 95 %DI—lower border of 95 % confidence interval, 95 %HI—upper border of 95 % confidence interval.

**Table 2 plants-11-00987-t002:** Correlation of deadwood volume and coverage with site characteristics determined by Pearson’s correlation coefficients.

Characteristics		Deadwood Volume (m^3^/ha)			Deadwood Coverage (%)			Volume of Fine Woody Debris (m^3^/ha)			Coverage of Fine Woody Debris (%)		
		r	*p*	Sign	R	*p*	Sign	r	*p*	Sign	r	*P*	Sign
Number of deadwood pieces		−0.0183	0.6714		0.4142	0.0000	***	0.4681	0.0000	***	0.6636	0.0000	***
Slope (%)		0.2344	0.0000	***	0.0717	0.0969	.	−0.1062	0.0138	*	−0.0980	0.0231	*
Aspect (Degrees)		0.2069	0.0000	***	0.1318	0.0022	**	0.0611	0.1573		0.0070	0.8711	
Vegetation coverage (%)		−0.1435	0.0009	***	−0.2074	0.0000	***	−0.0946	0.0283	*	−0.1915	0.0000	***
Plant litter coverage (%)		−0.1854	0.0000	***	−0.0028	0.9477		0.1320	0.0022	**	0.1288	0.0028	**
Micro-relief	v v	0.1482	0.0006	***	0.1164	0.0069	**	0.0461	0.2866		0.0324	0.4539	
	v —	−0.0300	0.4874		−0.0167	0.6999		0.0175	0.6851		0.0054	0.9008	
	v ^	0.0999	0.0206	*	0.0465	0.2822		−0.0483	0.2634		−0.0515	0.2332	
	– v	0.0477	0.2699		−0.0222	0.6085		−0.0544	0.2080		−0.0511	0.2376	
	– –	−0.1430	0.0009	***	−0.0515	0.2338		0.0445	0.3039		0.0673	0.1195	
	– ^	0.0053	0.9016		−0.0060	0.8889		−0.0292	0.4989		−0.0412	0.3409	
	^ v	−0.0269	0.5340		−0.0496	0.2516		−0.0275	0.5247		−0.0284	0.5115	
	^ –	0.0007	0.9871		−0.0248	0.5666		−0.0466	0.2809		−0.0468	0.2793	
	^ ^	0.0208	0.6308		−0.0187	0.6647		−0.0534	0.2168		−0.0632	0.1434	
Vegetation distribution	Clustered	0.0895	0.0381	*	0.1559	0.0003	***	0.1092	0.0113	*	0.1481	0.0006	***
	No vegetation	0.0100	0.8170		0.1382	0.0013	**	0.0880	0.0416	*	0.1693	0.0001	***
	Regular	−0.0916	0.0338	*	−0.1809	0.0000	***	−0.1251	0.0037	**	−0.1786	0.0000	***
Deadwood distribution	Clustered	−0.2524	0.0000	***	−0.1120	0.0094	**	0.0415	0.3375		0.0493	0.2544	
	No deadwood	−0.0408	0.3454		−0.1301	0.0025	**	−0.1271	0.0032	**	−0.1425	0.0009	***
	Regular	0.3836	0.0000	***	0.2766	0.0000	***	0.0640	0.1386		0.0680	0.1157	
Humus form	No humus	0.0782	0.0700	.	−0.0902	0.0367	*	−0.1342	0.0018	**	−0.1457	0.0007	***
	Moder	0.0123	0.7762		0.1165	0.0069	**	0.1438	0.0008	***	0.1202	0.0053	**
	Mull	−0.0540	0.2118		−0.0619	0.1520		−0.0640	0.1384		−0.0354	0.4129	

Legend: r—Pearson correlation coefficient, *p*—critical *p* value. Significance: 0 ‘***’ 0.001 ‘**’ 0.01 ‘*’ 0.05 ‘.’ 0.1. Microrelief (- flat land, ^ concave terrain, v convex terrain, along the contour line and along the slope).

**Table 3 plants-11-00987-t003:** Statistical characteristics of models quantifying deadwood volume (m^3^/ha) using a function f(x) with predictors: Coverage represents the proportion of plot area covered by all deadwood pieces if placed next to each other given in %, and Diameter is mean diameter of deadwood pieces in mm.

No.	Model	R^2^	R^2^ _adj_	F	*p*	AIC	BIC
1	Volume (m^3^/ha) = f(Coverage)	0.42	0.42	392.13	0.00	7631.11	7643.97
2	Volume (m^3^/ha) = f(Diameter)	0.43	0.43	406.09	0.00	7623.09	7635.95
3	Volume (m^3^/ha) = f(Coverage + Diameter)	0.63	0.63	460.75	0.00	7389.91	7407.06
4	Volume (m^3^/ha) = f(Coverage × Diameter)	0.83	0.83	858.15	0.00	6983.62	7005.05
5	Volume (m^3^/ha) = f(poly(Coverage, 2))	0.57	0.57	358.88	0.00	7470.90	7488.04
**6**	**Volume (m^3^/ha) = f(polym(Coverage, Diameter, degree = 2))**	**0.91**	**0.91**	**1065.99**	**0.00**	**6644.87**	**6674.87**

Legend: R^2^—coefficient of determination, R^2^_adj_—adjusted coefficient of determination, F—test statistics of Fisher–Snedecor distribution, *p*—critical *p* value of F-statistics, AIC—Akaike information criterion, BIC—Bayesian information criterion.

**Table 4 plants-11-00987-t004:** Statistical parameters of the best model (6) from Table 3 describe the relationship of Deadwood volume (m^3^/ha) = f(x). Coverage represents the proportion of a plot area covered by all deadwood pieces if placed next to each other given in %, and Diameter is mean diameter of deadwood pieces in mm.

	Predictor	Regression Coefficient	SE	t	p_t_	Sign	R^2^	R^2^_adj_	F	Df	p_F_
1	Intercept	−5.227837	11.40	−0.46	0.647		0.9094	0.9085	1066	5/531	<2.2 × 10^−16^
2	Coverage	−6.406274	0.92	−6.95	0.000	***					
3	Coverage ^2	0.158176	0.02	9.78	0.000	***					
4	Diameter	1.545334	0.36	4.34	0.000	***					
5	Coverage × Diameter	0.185188	0.01	26.32	0.000	***					
6	Diameter ^2	−0.009702	0.00	−17.55	0.000	***					

SE—standard error, t—t value of Student’s t-test, pt—*p* value of t value, Sign—significance of *p* value as follows: *** 99.9%, R^2^—R squared, R^2^_adj_—adjusted R-squared, F—test statistics of Fisher–Snedecor distribution, Df—degrees of freedom, pF—*p* value for F.

**Table 5 plants-11-00987-t005:** Statistical characteristics of models quantifying the volume of fine woody debris (FWD): VolumeFWD(m^3^/ha) = f(x). Coverage represents the proportion of a plot area covered by all deadwood pieces if placed next to each other given in %, and Diameter is mean diameter of deadwood pieces in mm.

No.	Model	R^2^	R^2^_adj_	F	*p*	AIC	BIC
1	VolumeFWD(m^3^/ha) = f(Coverage)	0.89	0.89	4228.38	0.00	3869.45	3882.30
2	VolumeFWD (m^3^/ha) = f(Diameter)	0.09	0.09	52.77	0.00	4993.06	5005.91
3	VolumeFWD (m^3^/ha) = f(Coverage + Diameter)	0.92	0.92	3025.43	0.00	3696.55	3713.70
4	VolumeFWD (m^3^/ha) = f(Coverage × Diameter)	0.96	0.96	4339.76	0.00	3309.95	3331.38
5	VolumeFWD (m^3^/ha) = f(poly(Coverage, 2))	0.89	0.89	2164.03	0.00	3859.43	3876.58
6	VolumeFWD (m^3^/ha) = f(polym(Coverage, Diameter, degree = 2))	0.96	0.96	2608.25	0.00	3311.14	3341.14
**7**	**VolumeFWD (m^3^/ha) = f(Coverage + (Coverage × Diameter))**	**0.96**	**0.96**	**6520.46**	**0.00**	**3308.06**	**3325.20**

Legend: R^2^—coefficient of determination, R^2^_adj_—adjusted coefficient of determination, F—test statistics of Fisher–Snedecor distribution, *p*—critical *p* value of F statistics, AIC—Akaike information criterion, BIC—Bayesian information criterion.

**Table 6 plants-11-00987-t006:** Statistical parameters of the best model, No. 7 from Table 5, describe the relationship Volume of fine woody debris (m^3^/ha) = f(x). Coverage represents the proportion of a plot area covered by all deadwood pieces if placed next to each other given in %, and Diameter is mean diameter of fine deadwood pieces in mm.

	Predictor	Regression Coefficient	SE	t	|p_t_|	Sign	R^2^	R^2^_adj_	F	Df	p_F_
1	Intercept	−1.596314	0.33	−4.77	0.00	***	0.9607	0.9605	6520	2/534	<2.2 × 10^−16^
2	Coverage	1.240540	0.06	21.48	0.00	***					
3	Coverage × Diameter	0.070155	0.00	31.48	0.00	***					

SE—standard error, t—t value of Student’s t-test, pt—*p* value of t value, Sign—significance of *p* value as follows: *** 99.9%, R^2^—R squared, R^2^_adj_—adjusted R-squared, F—test statistics of Fisher–Snedecor distribution, Df—degrees of freedom, pF—*p* value for F.

**Table 7 plants-11-00987-t007:** Basic characteristics of disturbed areas and respective forest stands prior to the disturbance.

Area No.	Forest Stand Prior to Disturbance			Slope (%)	Elevation Min-Max (m a.s.l.)	Disturbed Area (ha)	Perimeter of Disturbed Area (m)	Sampling Intensity	Number of Sample Plots (pcs)
	Tree species/share (%)/site index (m)	Age (yrs)	Stocking						
1	*Abies alba* Mill./7/30; *Acer pseudoplatanus* /1/26; *Fagus sylvatica*/80/28; *Fraxinus excelsior*/1/30; *Picea abies*/11/32	115	0.7	50	550–650	5.29	1153.49	0.002	110
2	*Abies alba* Mill./15/40; *Fagus sylvatica*/77 /38; *Picea abies*/2/38; *Quercus petraea*/6/36	85	0.9	35	510–620	2.44	1263.71	0.002	51
3	*Abies alba* Mill./15/40; *Fagus sylvatica*/80/38; *Quercus petraea*/5/36	85	0.9	30	530–620	0.75	454.43	0.004	31
4	*Abies alba* Mill./10/32; *Fagus sylvatica*/85/34; *Quercus petraea*/5/28	85	0.85	40	485–585	4.30	1454.48	0.002	93
5	*Abies alba* Mill./5/34; *Fagus sylvatica*/90/33; *Quercus petraea*/5/30	85	0.9	30	550–660	12.59	2509.54	0.002	251
Sum						25.38			536

## Data Availability

The data presented in this study are available from the corresponding author on request. The data are not publicly available due to the ongoing analyses.

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
