# Peer review of "Deadwood Amount at Disturbance Plots after Sanitary Felling"

_plants, 2022, doi:10.3390/plants11070987_

Round 1
Reviewer 1 Report
The subject is out of Plants scope. It is rather for Forestry journals.
Author Response
Dear Reviewer,
Thank you very much for the review of our paper. Below please find our responses to your comments.
The subject is out of Plants scope. It is rather for Forestry journals
Thank you for the comment, we will reconsider it if the manuscript is not accepted in Plants.
Reviewer 2 Report
The article deals with a very important research topic.
The title is consistent with the content of the article.
The abstract is correctly written.
The introduction is well written.
The results are described well, good charts are discerned with very good statistical analysis.
The discussion is substantive and refer to the research results of other researchers.
Conclusion are very good.
The literature is well chosen.
I see one small error in the text: line 99? the text "(Error! Reference source not found.)"
This is a very good article, congratulations!
Author Response
Dear Reviewer,
Thank you very much for the review of our paper. Below please find our responses to your comments.
I see one small error in the text: line 99? the text "(Error! Reference source not found.)
Thank you, we corrected it.
Thank you for the positive feedback. We highly appreciate it.
Reviewer 3 Report
Dear Authors,
please, find below several remarks regarding your manuscript:
- general request: please, change the position of the „4. Materials and methods” section to pos. 2, directly after „1. Introduction”
- lines 64-67: please, move these sentences to the „Materials and methods” section
- line 108 – please explain the definition of the „deadwood coverage” variable in the methodology section (how it was calculated?)
Impressive literature review.
Good statistical analyses.
Best regards!
Author Response
Dear Reviewer,
Thank you very much for the review of our paper. Below please find our responses to your comments.
general request: please, change the position of the „4. Materials and methods” section to pos. 2, directly after „1. Introduction”
unfortunatelly, this is the structure required by Plants, so we cannot change it.
lines 64-67: please, move these sentences to the „Materials and methods” section
We are not sure if we uderstand the comment correctly, as the text in lines 64 to 67 states the hypothesis. Since hypotheses are usually given together with goals, we did not move the text to Methods as suggested by the reviewer, but tried to reformulate the text to clarify our intention. We hope it is now acceptable.
line 108 – please explain the definition of the „deadwood coverage” variable in the methodology section (how it was calculated?)
We added this information in Methods
We hope the changes are acceptable for you.
Thank you for the positive feedback. We highly appreciate it.
Round 2
Reviewer 1 Report
This is not Plant Ecology, and I cannot suggest publication in PLANTS. As I mentioned previously no plants in focus. This is forestry and forest management and economy paper. Also not for Plants readerships.
Author Response
Thank you for the comment, we believe this is upon the decision of the editorial board. The editorial board contacted us before the submission of the paper and chose the paper based on the abstract.
Reviewer 3 Report
Dear Authors,
thank you for your response, explanations, and changes applied to the manuscript.
I found you have added the information about the conditions of the site, where you collected your samples. In the first version of the manuscript, this information has been given in lines 346-354. In the revised version of the manuscript, almost the same information (with a small difference in the first sentence) has been given in lines 369-377 and 383-390.
I'm not sure it is necessary for such repetition. Maybe you can keep the lines 369-377 and in next, you can just make the reference to the description mentioned above (lines 369-377)?
Best regards!
Author Response
Thank you for the comment. There was a mistake in pdf due to the tracked changes. We deleted the second part from the text.